# Prevalence and predictors of secondary traumatic stress symptoms in health care professionals working with trauma victims: A cross-sectional study

**Nina Ogińska-Bulik**[1], **Piotr Jerzy Gurowiec**[2], **Paulina Michalska**[1]*, **Edyta Kędra**[3]

**1** Department of Health Psychology, Institute of Psychology, University of Lodz, Lodz, Poland, **2** Institute of Health Sciences, University of Opole, Opole, Poland, **3** Medical Institute, State Higher Vocation School in Glogow, Glogow, Poland

* paulina.michalska@now.uni.lodz.pl

## Abstract

**Data Availability Statement:** All relevant data are within the paper and its Supporting information files.

### Introduction

Medical personnel is an occupational group that is especially prone to secondary traumatic stress. The factors conditioning its occurrence include organizational and work-related factors, as well as personal features and traits. The aim of this study was to determine Secondary Traumatic Stress (STS) indicators in a group of medical personnel, considering occupational load, job satisfaction, social support, and cognitive processing of trauma.

### Material and methods

Results obtained from 419 medical professionals, paramedics and nurses, were analyzed. The age of study participants ranged from 19 to 65 ($M = 39.60$, $SD = 11.03$). A questionnaire developed for this research including questions about occupational indicators as well as four standard evaluation tools: Secondary Traumatic Stress Inventory, Job Satisfaction Scale, Social Support Scale which measures four support sources (supervisors, coworkers, family, friends) and Cognitive Processing of Trauma Scale which allows to evaluate cognitive coping strategies (positive cognitive restructuring, downward comparison, resolution/acceptance, denial, regret) were used in the study.

### Results

The results showed that the main predictor of STS symptoms in the studied group of medical personnel is job satisfaction. Two cognitive strategies also turned out to be predictors of STS, that is regret (positive relation) and resolution/acceptance (negative relation). The contribution of other analyzed variables, i.e., denial, workload and social support to explaining the dependent variable is rather small.

**Funding:** The study was supported by University of Opole (Poland) internal grant "Application for funding a research project under a grant for maintaining research potential in 2020 - WPBIN 1/19". The funders had no role in study design, data collection and analysis, decision to publish, or preparation of the manuscript.

**Competing interests:** The authors have declared that no competing interests exist.

## Conclusions

Paramedics and nurses are at the high risk of indirect traumatic exposure and thus may be more prone to secondary traumatic stress symptoms development. It is important to include the medical personnel in the actions aiming at prevention and reduction of STS symptoms.

## Introduction

People who professionally help trauma victims are indirectly exposed to it themselves. In the fifth edition of the Diagnostic and Statistical Manual of Mental Disorders (DSM-5), the etiological factor (experience of a traumatic event) was extended to include indirect exposure to a traumatic event [1].

Secondary trauma concern professionals who provide assistance to trauma victims and sufferers. A special place among them has the representatives of the medical personnel who are often the first to contact trauma victims [2, 3]. Witnessing the death of patients as well as the necessity to conduct difficult conversations with patients and their families causes stress and negative emotions as well as their suffering in many of them. Exposure to indirect trauma may be connected with various mental health disorders, above all Secondary Traumatic Stress (STS), which is also described as Secondary Traumatic Stress Disorder (STSD). Some authors describe that STS, defined as stress resulting from helping or willing to help individuals experiencing trauma, may lead to secondary post-traumatic stress disorder; persistence of high STS symptoms allows for the diagnosis of STSD [3, 4].

The notion of secondary traumatic stress was popularized by Charles Figley [3] who described it as stress connected with helping other suffering people or trauma victims. It is defined as the behavioral and emotional outcomes experienced by an individual upon gaining knowledge of another person's stressful experiences [3, 4]. The introduction of secondary traumatic stress was preceded by the notion of compassion fatigue [3]. It was initially used in relation to nurses and then broadened to therapists and other professionals dealing with the mental health of people who were exposed to traumatic events. Figley [3] stressed that these professionals are the first to ease the pain and suffering of people who experienced trauma. Yet, while helping others they also become trauma victims. Another term used in relation to the discussed phenomenon is the Vicarious Traumatization (VT). This term was introduced by McCann and Pearlman [5] to describe the changes in the therapist's worldview which occur as a result of empathic engagement in helping patients who experienced trauma. Vicarious traumatization means the transformation of the internal experience of helpers resulting from their therapeutic work for the client. This concept, as underlined by Tosone et al. [6] has a slightly softer overtone than secondary traumatic stress but most often these terms are treated as synonyms.

STS symptoms are related to thoughts, emotions, and behaviors resulting from the knowledge about traumatic events experienced by others but also from engagement in helping the trauma victims. They include the same symptoms which occur in PTSD and are experienced by people directly exposed to trauma [3, 7]. Therefore STSD may also be referred to as secondary PTSD [3]. According to the new DSM-5 classification [1], the occurring symptoms belong to the fourth category, which is intrusion, avoidance, negative alterations in cognitions and mood, and alterations in arousal and reactivity.

### STS among medical personnel

Medical personnel representatives—nurses, doctors, paramedics, that is people who have direct contact with patients and those who suffer because of their conditions or injured in

accidents—are subject to negative consequences of exposure to trauma [8–13]. As underlined by Beck [14], STS is perceived as a professional risk factor among health care professionals. Research carried out within this scope confirms the high risk of STS occurrence in this professional group.

The studies conducted among paramedic personnel, the majority of which consisted of doctors, in 10 hospitals of one of the States in the USA shows that nearly 13% meet the STS criteria and almost 34% presented at least one symptom included in the scope of each of three STS categories, that is an intrusion, arousal and avoidance [15]. One of the studies mentioned by Nimmo and Huggard [12] shows that more than half (54%) of doctors who participated in the study met the criteria of compassion fatigue or STS. Yet, it results from other studies presented by the authors that the intensity of STS among doctors was low. Symptoms of secondary traumatic stress were also revealed by ambulance personnel in the study of Argentero and Setti [16].

Research carried out among nurses working in emergency rooms in Scotland shows that 75% of them presented at least one symptom included in the STS scope [17]. According to studies [18], 64% of Irish nurses working in emergency medical services met the STS criteria Similarly, 52.3% of emergency nurses in Jordan revealed a high or severe level of secondary traumatic stress [19]. High risk of the occurrence of secondary traumatization in this professional group is confirmed by study results which show that 86% of nurses participating in the study presented a moderate or high level of compassion fatigue [20]. Other studies mention frequent STS symptoms occurrence among oncology and critical care nurses [21–23]. High risk of STS symptoms occurrence is also observed among nurses who work in palliative care and cancer wards. Nurses who deal with the seriously ill, wounded, and those facing the end-of-life over extended periods of time are at particular risk of negative effects of indirect trauma [11, 20, 24]. It is confirmed by the literature review by Ortega-Compos et al. [25] which shows that 19% of cancer ward nurses present low compassion satisfaction, 56% moderate or high level of burnout, and 60% moderate or high level of compassion fatigue.

Polish studies confirm the high risk of STSD occurrence among the medical personnel. In research concerning nurses working in palliative care [26] escalation of STS symptoms (measured by means of Secondary Traumatic Stress Scale) was reported in 38.9% of participants. Moderate STS symptoms were observed in 23.6% and 37.5% of participants showed low levels of STS symptoms. In studies in which Posttraumatic Checklist—PCL-5 was applied to evaluate STS symptoms [13] it was reported that among 5 professional groups (therapists, paramedics, nurses, social workers, and probation officers) the highest level of STS symptoms was presented by medical personnel and high probability of STSD occurrence was noted in 45.8% of paramedics and 40% of nurses employed in posttraumatic and palliative care wards.

## Professional load, work satisfaction, social support and cognitive trauma processing versus secondary traumatic stress

Several theoretical models have been developed to explain the nature of secondary traumatization and describe the factors that determine the occurrence of secondary traumatic stress. One of the most important concept that directly relates to secondary trauma is Ecological Framework of Trauma by Dutton and Rubinstein [27]. The authors distinguished several elements of the model such as: (1) traumatic events experienced by the victim, (2) coping strategies understood as cognitive and behavioural efforts undertaken by the helper, (3) posttraumatic reactions of the helper, (4) subjective factors that include personal resources (especially high self-esteem), professional resources (experience, training), vulnerability (own trauma history), satisfaction levels, both in life and at work, and environmental factors that include social support

and working environment; the context of trauma worker works. The authors paid attention mostly to personal and environmental factors. These factors influence the emotional reactions of those helping people working with trauma victims that may be reflected as secondary traumatic stress. Coping strategies are also important, due to the fact that coping activity is aimed to master the requirements of an individual's traumatic situation. Another notable concept is Constructivist self-development theory by McCann and Pearlman linked to previously described phenomenon of Vicarious Traumatization [5]. The theory evokes changes in cognitive patterns or beliefs, and therefore may be relevant to secondary trauma. It is worth mention that models developed for PTSD, i.e., the Cognitive shattered assumptions theory by Janoff-Bulman [28], PTSD Development model [29] and Emotional processing model [30] may also be used to explain mechanisms of secondary traumatization. The applicability of these models to STS symptoms results primarily from symptomatic similarity between STS and PTSD (as indicated by Figley) and analogical factors that influence the occurrence of these two-side consequences of trauma. These models treat posttraumatic disorders as errors in cognitive trauma processing which leads to the occurrence of distorted beliefs concerning the world and self. Such distortions entail intensive emotional reactions, mostly in the form of anxiety and foster the occurrence and sustainment of trauma symptoms.

Taking into account the assumptions of above models and studies conducted in the field of secondary trauma [13, 31, 32], it can be pointed out that special attention is focused on factors connected with work environment including occupational load, job satisfaction, social support as well as individual characteristic including cognitive trauma processing skills.

**Occupational load characteristic.**   Steed and Bicknell [32] report the fact that occupational load, especially in the form of numerous patients and a long time devoted to working with them is the main environmental risk factor of secondary trauma occurrence. Yet, the conducted study does not give an explicit image of connections between variables. Studies related to various professional groups, i.e. therapists [32], trauma counselors [33], other professionals [34] show positive links between occupational load and STS symptoms. Research carried out among nurses [35] also provided data pointing toward the positive link between occupational load and STS symptoms. Positive links between occupational load in the form of e.g., years of practice and STS were also revealed by emergency nurses [36, 37]. Similar, years of work experience found to be positively associated with the level of compassion fatigue among oncology and critical care nurses [38, 39] and the group of paramedics [40]. Hinderer et al. [37] found that the number of hours worked per shift was associated with greater STS. Yoder [41] stressed that heavy workload causes secondary traumatic stress among nurses from different hospital wards, i.e. emergency unit, oncology unit, or intensive care unit.

Other data do not confirm the connection between occupational characteristic and secondary traumatization symptoms [42–44]. Similarly, the study by Duffy, Avalos, and Dowling [18] conducted among emergency nursing did not confirm the link between STS and workload and experience. Baird and Jenkins [45] indicate that those seeing more trauma clients reported less distress. In the study of Mooney et al. [46] oncology and intensive care nurses with more years of experience revealed a lower level of compassion fatigue than nurses with less experience. This issue required further analyses.

**Job satisfaction.**   Few studies were conducted in relation to links between job satisfaction and STS symptoms. The majority of them are focused on satisfaction from helping to treat it as opposed to compassion fatigue [47] and not on the general feeling of work satisfaction. The negative connections of satisfaction from helping with STS symptoms, compassion fatigue, and burnout were presented in the study of American nurses [37]. Low job satisfaction, more work hours, and second-hand smoke exposure were related to secondary traumatic stress, explaining 9% of the variance in nurses from the central part of China [35]. Another study

[48] showed that nurses more satisfied with the job reported a lower level of compassion fatigue than their less satisfied colleagues. Kelly and Lefton [49] indicated that job satisfaction played the role of predictor for STS; reduce the severity of secondary traumatic stress symptoms and increase compassion satisfaction among critical care nurses.

The negative connection between work satisfaction and STS symptoms was revealed in other professional groups as well. Studies conducted on consultants working with substance addicts serve a good example [50]. It was also reported in it that work satisfaction mediated the link between STS and engagement in work. In Polish research of professionals who help trauma victims [13] job satisfaction was not connected to STS symptoms. Similar results were obtained in the study of Balinbin et al. [51] among nurses. Some research [52] indicated inverse relationships; secondary traumatic stress leads to job dissatisfaction. The scarcity of research on this topic and ambiguous results point to the need for further studies.

**Social support from the work environment.** Social support understood as help available for an individual in stressful situations [53] may serve a protective role in the process of secondary traumatization. The available data do not provide a clear view of the dependence between social support and the negative consequences of secondary trauma. It is connected with the ambiguity of the social support construct, its types (perceived obtained), kinds (emotional, informational, instrumental) and sources (spouse, parent, colleague, supervisor).

In case of specialists working with trauma victims, it seems that the particular role is played by support of the work environment, that is supervisors and coworkers. It can reduce stress levels and influence the levels of experienced emotions—increasing the positive and lowering the negative ones—as well as correct distorted cognitive schemes [54]. From research conducted among Chinese oncology nurses results that support from organization was identified as significant protector of compassion fatigue [39]. The obtained support may, therefore, be the factor protecting against the development of STS.

Manning-Jones et al. [31] present data showing that 95% of examined professionals working with trauma victims engage in peer support and 58% declare that they were supported by supervisors. Moreover, in the group of nurses, the support provided by colleagues or coworkers was the main negative STS predictor. Research conducted with nurses working in emergency rooms [18] and intensive care [55] units showed that support from coworkers plays a significant role in alleviating STS symptoms. In a study by Jonsson and Halabi [56] lack of support in the work environment was connected with the occurrence of STS.

Research conducted with rescue workers, involved in critical operations of various kinds in constant contact with traumatized subjects [11] provided data that shows that work-related factors, especially support in the work environment are connected with the occurrence of STS symptoms. In Polish research [13] STS symptoms correlated negatively with the support obtained from coworkers in the group of therapists. Yet, no such relation was reported in the group of paramedics and nurses.

Other studies showed a significant role of support obtained from relatives and friends in alleviating the negative consequences of secondary traumatization among nurses [57, 58]. Similarly, Von Rueden et al. [59] underline that the nurses who obtained social support from relatives and friends experienced STS symptoms less frequently in comparison to those for whom such support was not provided. In Poland, there is no studies of nurses which shows the relations between support from their friends and family and STS symptoms.

**Cognitive trauma processing.** According to the previously mentioned models and theories, the significant role in the development of STS is played by cognitive trauma processing [5, 13, 27–30]. Cognitive trauma processing may be mirrored in the undertaken cognitive countermeasures. Their aim is to give meaning to the experienced and adjustment to the new reality, changed as a result of experienced trauma. Williams, Davis and Millsap [60] list several

factors as indicators of effective trauma processing, including a decrease in the level of negative emotions (especially feelings of guilt or shame), the assimilation of information about the traumatic event, the acceptance of the event, the perception of its positive aspects, and desensitization, manifested as a gradual reduction of the perceived stress and negative emotions associated with ruminating on the event. Such cognitive trauma processing is often realized through cognitive coping strategies in the form of positive cognitive restructuring, downward comparison, resolution/acceptance, regret and denial.

Research including correctional psychologists [61] provided data pointing towards the occurrence of the negative relations of beliefs relating to both the goodness of the surrounding world and its comprehensibility to STS symptoms. In the study of five groups of professionals working with trauma victims (therapists, paramedics, nurses, social workers, and probation officers) a significant role of cognitive trauma processing in the occurrence of STS symptoms was confirmed [13]. Positive links between STS symptoms and interference in core beliefs (evaluated by means of Core Beliefs Inventory), rumination about the traumatic events experienced by patients (evaluated by means of Event Related Rumination Inventory) as well as negative coping strategies (evaluated by means of Cognitive Processing of Trauma Scale), that is regret and denial were confirmed. Few studies of relations between cognitive trauma processing and negative consequences of secondary exposure to trauma refer directly to medical personnel. Results of research conducted among paramedics [10] provided data pointing towards the fact that dysfunctional beliefs and dysfunctional strategies of coping with intrusion played a predictive role for PTSD and STS symptoms.

**Aim of the study.**   The adopted research model refers mainly to Ecological Framework of Trauma by Dutton and Rubinstein [27] which includes four elements: (1) traumatic events experienced by the victim, (2) coping strategies undertaken by the helper, (3) posttraumatic reactions of the helper, (4) subjective and environmental factors, including personal resources, work environment, job satisfaction, and social support.

The undertaken research aimed at indicating the determinants of secondary traumatic stress symptoms among medical personnel exposed to trauma experienced by their patients. The determinants included environmental and work-related variables, i.e., occupational load expressed in the form of three indicators, that is work experience as paramedic/nurse, number of hours per week devoted to helping patients as well as workload expressed by the proportion of work devoted to direct help for patients in reference to the entire performed work. Work-related and environmental variables included also job satisfaction and social support provided by supervisors and coworkers. Moreover, the support provided by family and friends was also taken into consideration as well as individual factors, i.e., cognitive trauma processing in the form of five cognitive strategies of coping with trauma experienced by patients.

It was hypothesized in the study that STS symptoms will be positively connected with the occupational load indicators and negatively with job satisfaction and social support. It was also hypothesized that the main determinant of STS will be cognitive trauma processing and that negative strategies (regret, denial) will be positively linked with STS symptoms and positive strategies, especially resolution/acceptance and positive cognitive restructuring will be linked negatively.

## Materials and methods

### Participants

The research project was specifically approved by the Bioethics Committee of Opole Medical School (no 81/P1/2019). Informed consent (oral before, and written during filling the questionnaires) was obtained from all participants included in the study. 430 representatives of

medical personnel who provide medical help to trauma victims were included in the study (the sample consists of individuals who were exposed to secondary trauma). The questionnaires were delivered to medical staff who had previously consented (written) to participate in the project. The study was anonymous and voluntary, and conducted in the period from November 2019 to February 2020 in 12 units and included voivodeship rescue stations, emergency medical teams, emergency wards in several Polish hospitals as well as cancer wards, intensive care units, and hospices. The questionnaires were delivered to and collected by the authors or persons trained by the authors during medical staff working hours. The study inclusion criterion was performing the profession of a paramedic or nurse and work with people who had traumatic experiences (struggling with illness. i.e., stroke, heart attack, cancer or after an accident).

The analysis included the results of 419 (11 questionnaires were dropped out due to the missing data; only participants with complete questionnaires were included in the analyses) people in the age from 19 to 65 years ($M = 39.60$, $SD = 11.03$). Among study participants, there were 137 (32.7%) men and 282 (67.3%) women. The studied group included paramedics ($n = 201$) where 60.2% were men and nurse staff ($n = 218$) in which there was a significant majority of women (92.7%). The majority of paramedics provide help for people who experienced various accidents, especially road accidents (57.2) but also after traumatic events such as strokes and cardiac infarction (42.8%). Nurse staff included people working with cancer patients (87.7%) and accident sufferers (18.3%). Work experience of the medical personnel who participated in the study amounted at from 1 year to 43 years ($M = 12.18$, $SD = 9.75$), the number of work hours devoted to helping injured patients amounted at 2 to 90 ($M = 38.24$, $SD = 15.65$), and workload expressed by a percent of work devoted directly to providing help to patients in relation to the whole performed work from 2 to 100% ($M = 69.11$, $SD = 31.89$).

## Measures

The survey developed for the research was used. It included questions about age, types of events which were experienced by patients, work experience as a paramedic/nurse, number of work hours per week devoted to working with patients, workload expressed in the percentage of work devoted to providing direct help for patients in relation to the whole performed work as well as four standard assessment tools described below.

**Secondary Traumatic Stress Inventory—STSI** is a modified version of the Posttraumatic Stress Disorder Checklist—PCL-5 developed by Weathers et al. [62]. The inventory is a self-assessment tool intended for testing people who provide help for trauma victims. Similarly to PCL-5 which was adapted to Polish conditions [63], it consists of 20 statements/accounts of traumatic events (range = 0–80; *"Repeated, disturbing, and unwanted memories of the stressful experience"*) referring to symptoms included to 4 criteria of PTSD, that is B. Intrusion; C. Persistent avoidance of stimuli connected to trauma; D. negative alterations in cognitions and mood; E. alterations in arousal and reactivity. The modification of the tool consisted of completing the instructions with the information about the occurrence of mentioned reactions in connection to the help provided to trauma victims. Some statements were completed with the phrase "of my patients". According to the instruction the study participant shows to what extent the mentioned reactions occurred to them within last month in connection to the provided help and evaluates it using a five-level scale from *not at all* (0); *slightly* (1); *moderately* (2); *significantly* (3) to *very much* (4). Cronbach's *alpha* indicator for Secondary Traumatic Stress Inventory amounts at 0.90 and is following for particular factors: 0.71; 0.85; 0.89; 0.87.

**Job Satisfaction Scale** is a modified version of Diener's Satisfaction with Life Scale [64], designed to evaluate the general life satisfaction, developed by Zalewska [65]. The tool consists

of 5 items. After the alterations, the statements refer to the evaluation of work *("In many aspects my work is almost perfect")*. The study participant uses a seven-point scale of responses from 1 –*"I certainly agree"* to 7 –*"I certainly disagree"*. All statements are a part of one dimension (range = 7–35) and are internally highly consistent in a heterogeneous sample of employees and particular professional groups. Cronbach's *alpha* for scale is 0.86.

**Social Support Scale—What Support Can You Expect** is a part of the Psychosocial Work Conditions questionnaire [66] and allows to evaluate the support received from the work environment, i.e. supervisors and coworkers as well as support outside of work, i.e. from family and friends (score range for each subscale = 8–40; *"To what extent can you expect that someone helps you in a certain way?")*. The tool consists of 8 statements for which participants answer on a 5-point scale from 1 *(very small extent)* to 5 *(very large extent)*. Psychometric properties of the scale are satisfactory (support from supervisors: $\alpha = 0.94$, coworkers: $\alpha = 0.92$, friends outside of work: $\alpha = 0.89$ and family: $\alpha = 0.89$).

**Cognitive Processing of Trauma Scale** (CPOTS) by Williams, Davis and Millsap [60] was adapted to Polish conditions by Ogińska-Bulik and Juczyński [67]. A version adjusted for the study of people indirectly exposed to trauma was applied in the research. The tool consists of 17 statements *("Overall, there is more good than bad in this experience")* and measures five aspects of cognitive processing: positive cognitive restructuring (3 items; score range = 0–18), downward comparison (3 items; score range = 0–18), resolution/acceptance (4 items; score range = 0–24), denial (4 items; score range = 0–24), and regret (3 items; range = 0–18). Study participants address each statement on a seven-point scale from 0 (*I certainly disagree*) to 6 (*I certainly agree*). The result of each scale is calculated separately. The reliability of the Polish version of CPOTS evaluated by means of Cronbach's *alpha* coefficient is satisfactory. The coefficients are 0.84 for positive cognitive restructuring, 0.89 for downward comparison, 0.82 for resolution/acceptance, 0.56 for denial and 0.72 for regret.

## Statistical analyses

The IBM SPSS, version 25 software was used to verify the obtained data. The two-tailed probability value of < 0.05 was considered to be statistically significant. The first step of data analysis consisted of the calculation of descriptive statistics that included mean and standard deviation for secondary traumatic stress, occupational characteristic, job satisfaction, source of social support and cognitive coping strategies. Additional for demographic characteristic frequency and percentage were computed. T-Student's test was implemented to compare the differences in the prevalence of secondary traumatic stress between nurses and paramedics as well as between men and women. Pearson's correlation coefficients were applied to analyze the relations between the variables. The Benjamini-Hochberg procedure was done for multiple comparisons. A multivariable stepwise regression analysis was used in order to find a dependent variable (STS) predictors among independent variables (occupational load characteristic, job satisfaction, cognitive trauma processing, and social support). To assess model fit, $R^2$ was used. Moreover, regression analysis provided data that include adjusted $R^2$, $R^2$-changes, standardized regression coefficient ($\beta$), unstandardized regression coefficient (*B*), *F*-statistic, confident intervals for *B*, and *p*-value. Multicollinearity was checked by using tolerance (> 0.10) and variance inflation factor (< 5). The effect size for multiple regression analysis was > 0.35.

## Results

The intensity of STS symptoms in the studied group of medical personnel who provides help for sufferers (Table 1) is higher than in standardization tests [63] which included people who directly experienced various traumatic events (M = 26.0, SD = 18.66, p<0.001). Considering

**Table 1. Descriptive statistics and correlation coefficients among analyzed variables (N = 419).**

| Variables | 1 | 2 | 3 | 4 | 5 | 6 | 7 | 8 | 9 | 10 | 11 | 12 | 13 | 14 | 15 | 16 | 17 | 18 |
|---|---|---|---|---|---|---|---|---|---|---|---|---|---|---|---|---|---|---|
| 1. STS total | - | | | | | | | | | | | | | | | | | |
| 2. STS intrusion | 0.908*** | - | | | | | | | | | | | | | | | | |
| 3. STS avoidance | 0.832*** | 0.771*** | - | | | | | | | | | | | | | | | |
| 4. STS negative alternations in cognitions and mood | 0.963*** | 0.815*** | 0.759*** | - | | | | | | | | | | | | | | |
| 5. STS alterations in arousal and reactivity | 0.945*** | 0.785*** | 0.715*** | 0.886*** | - | | | | | | | | | | | | | |
| 6. Work experience | 0.083 | 0.117* | 0.106* | 0.053 | 0.066 | - | | | | | | | | | | | | |
| 7. Number of working hours | -0.208** | -0.224** | -0.198** | -0.205** | -0.157** | 0.085 | - | | | | | | | | | | | |
| 8. Workload | -0.119* | -0.126** | -0.099* | -0.121* | -0.092 | 0.137** | 0.540*** | - | | | | | | | | | | |
| 9. Job satisfaction | -0.401*** | -0.355*** | -0.322*** | -0.399*** | -0.376*** | -0.025 | 0.130** | 0.120* | - | | | | | | | | | |
| 10. SS supervisors | -0.103* | -0.092 | -0.088 | -0.097* | -0.100* | -0.110* | 0.011 | 0.083 | 0.355*** | - | | | | | | | | |
| 11. SS coworkers | -0.070 | -0.023 | -0.016 | -0.082 | -0.099* | -0.113* | -0.112* | -0.032 | 0.280*** | 0.618*** | - | | | | | | | |
| 12. SS family | -0.236** | -0.225 | -0.154** | -0.218** | -0.236** | -0.055 | -0.030 | -0.095 | 0.339*** | 0.214** | 0.397*** | - | | | | | | |
| 13. SS friends | -0.278*** | -0.313*** | -0.256*** | -0.251*** | -0.229** | -0.112* | 0.113* | 0.112* | 0.495*** | 0.471*** | 0.269*** | 0.491*** | - | | | | | |
| 14. CPOT positive cognitive restructuring | -0.170** | -0.150** | -0.098* | -0.184** | -0.156** | 0.032 | 0.154** | 0.074 | 0.355*** | 0.187** | 0.152** | 0.277*** | 0.316*** | - | | | | |
| 15. CPOT downward comparison | -0.040 | -0.047 | -0.045 | -0.017 | -0.050 | 0.026 | 0.101* | 0.076 | 0.180** | 0.083 | 0.024 | 0.150* | 0.206** | 0.587*** | - | | | |
| 16. CPOT resolution/ acceptance | -0.320*** | -0.289*** | -0.248*** | -0.310*** | -0.307*** | 0.031 | 0.180** | 0.132** | 0.405*** | 0.168** | 0.108* | 0.287*** | 0.352*** | 0.674*** | 0.510*** | - | | |
| 17. CPOT denial | 0.175** | 0.150** | 0.152** | 0.197** | 0.137** | 0.041 | 0.088 | 0.026 | 0.108* | 0.064 | -0.080 | 0.015 | 0.168** | 0.429*** | 0.627*** | 0.333*** | - | |
| 18. CPOT regret | 0.181** | 0.158** | 0.136** | 0.195** | 0.157** | 0.005 | 0.089 | 0.064 | 0.139** | 0.137** | -0.049 | -0.043 | 0.206** | 0.393*** | 0.474*** | 0.289*** | 0.695*** | - |
| Mean | 31.00 | 7.98 | 3.26 | 10.22 | 9.55 | 12.18 | 38.24 | 69.11 | 21.28 | 23.44 | 27.95 | 29.80 | 25.78 | 8.66 | 8.14 | 12.20 | 8.61 | 6.42 |
| Standard deviation | 19.59 | 4.93 | 2.28 | 7.35 | 6.51 | 9.75 | 15.65 | 31.89 | 6.65 | 8.38 | 7.38 | 7.062 | 8.42 | 4.23 | 4.43 | 5.46 | 5.22 | 4.15 |

STS = secondary traumatic stress; SS = Social support; CPOT = cognitive processing of trauma;

*$p < 0.05$;

**$p < 0.01$;

***$p < 0.001$. (two-tailed).

*$p < 0.05$ not significant after Benjamini-Hochberg correction for multiple comparisons.

33 points assumed as a cut-off point for the general STS result [13] it is reported that 237 people which constitutes 56.6% of study participants show low or moderate STS symptom levels. In turn, high levels of these symptoms signifying a high probability of STSD diagnosis occurred in 182 people which is 43.4% of study participants. The nurse staff representatives manifested slightly higher levels of STS symptoms ($M = 32.23$, $SD = 20.69$) in comparison to paramedics ($M = 29.67$, $SD = 18.28$), yet, this difference is not statistically relevant ($t(417) = -1.336$, $p > 0.05$). In both groups the percentage of people with high risk of STSD occurrence is similar; in the paramedics' group, it amounts to 43.3% and among nurses– 43.6%. Gender did not differentiate the level of STS symptoms (men: $M = 30.32$, $SD = 18.31$; women: $M = 31.33$, $SD = 20.20$; $t(417) = -0.496$, $p > 0.05$). Positive, although weak links occur between the age of study participants and STS ($r = 0.123$, p<0.05).

Using Pearson's correlation coefficients the links between variables in the entire research group were determined (Table 1).

It results from the data presented in Table 1 that STS symptoms are slightly connected with the occupational load indicators (the number of work hours per week $r = -0.208$, $p < 0.01$; workload $r = -0.119$, $p < 0.05$). Job satisfaction is found to be related significantly to STS total score ($r = -0.401$, $p < 0.001$) and all STS symptoms. It can be noticed that three sources of social support are connected with STS (family $r = -0.236$, $p < 0.01$; friends $r = -0.278$, $p < 0.001$; supervisors $r = -0.103$, $p < 0.05$). Cognitive trauma processing in the form of four out of five cognitive coping strategies is associated with the negative consequences of secondary exposure to trauma (regret $r = 0.181$, $p < 0.01$; denial $r = 0.175$, $p < 0.01$; positive cognitive restructuring $r = -0.170$, $p < 0.01$; resolution/acceptance $r = -0.320$, p < 0.001).

Then it was checked which of the included variables explained the predictive role of STS symptoms, considering general results and particular criteria as the variables to be explained. The obtained results are presented in Table 2.

**Table 2. Regression analysis for variables predicting STS (total score) in the examined group of medical staff ($N = 419$).**

| Predictors | Secondary Traumatic Stress | | | | | | |
|---|---|---|---|---|---|---|---|
| | $B$ | $BE$ | $\beta$ | $R^2$ | $T$ | $p$-value | 95.0% CI for $B$ |
| Job satisfaction | -0.844 | 0.145 | -0.286 | 0.16 | -5.818 | < 0.001 | -1.129; -0.559 |
| Resolution/acceptance | -0.989 | 0.170 | -0.276 | 0.06 | -5.805 | < 0.001 | -1.324; -0.654 |
| Regret | 1.019 | 0.267 | 0.216 | 0.06 | 3.813 | < 0.001 | 0.494; 1.544 |
| Denial | 0.685 | 0.215 | 0.182 | 0.02 | 3.184 | < 0.01 | 0.262; 1.108 |
| Number of working hours | -0.177 | 0.052 | -0.141 | 0.01 | -3.384 | < 0.01 | -0.279; -0.074 |
| SS friends | -0.264 | 0.113 | -0.114 | 0.01 | -2.344 | < 0.05 | -0.486; -0.043 |
| SS coworkers | 0.232 | 0.115 | 0.087 | 0.01 | 2.000 | < 0.05 | 0.004; 0.458 |
| Work experience | 0.172 | 0.082 | 0.086 | 0.01 | 2.089 | < 0.05 | 0.010; 0.344 |
| | | | | | | | |
| $F(8,410)$ | | | | | | | 25.920 |
| $R$ | | | | | | | 0.580 |
| $R^2$ | | | | | | | 0.336 |
| Adj.$R^2$ | | | | | | | 0.323 |
| $R^2$-changes | | | | | | | -0.002 |

$B$ = unstandardized regression coefficient; $BE$ = B error; $\beta$ = standardized regression coefficient; $t$ = $t$-test value; $p$ = the level of significance (two-tailed); 95.0% $CI$ = confident intervals; $R$ = correlation coefficient; $R^2$ = determination coefficient; Adj.$R^2$ = adjusted $R^2$. Cohen's $f^2 = 0.51$
Tolerance range from 0.493 to 0.967; VIF range from 1.053 to 2.528.

Thirteen variables were introduced into the regression model, i.e., work experience, number of working hours, workload, job satisfaction, social support from supervisors, coworkers, family and friends, and cognitive coping strategies in the form of positive cognitive restructuring, downward comparison, resolution/acceptance, regret, denial. Data presented in Table 2 show that finally eight variables entered the regression equation and explained almost 34% variance of the dependent variable (negative consequences of exposure to trauma). The main predictor of STS symptoms was job satisfaction ($\beta$ = -0.286, $p < 0.001$) which explains the most, that is 16% of the variance of the general STS result. A negative relation between variables occurs. This means that the bigger satisfaction with work, the lower levels of STS symptoms. Cognitive coping strategies such as regret ($\beta$ = 0.216, $p < 0.001$) or resolution/acceptance ($\beta$ = -0.276, $p < 0.001$) explain 6% each. Regret strategy presents a positive relation while resolution/acceptance, a negative one. The share of other variables amounts to less than 2%.

Satisfaction with work was also the main predictor of symptoms included in four STSD criteria (Table 3). For intrusion the predictive role was played mainly by resolution/acceptance ($\beta$ = -0.236), job satisfaction ($\beta$ = -0.223) and regret ($\beta$ = 0.208). Avoidance is mostly explained by the satisfaction with work ($\beta$ = -0.225), denial ($\beta$ = 0.223) and resolution/acceptance ($\beta$ = -0.170). Cognitive and emotional alterations were explained by job satisfaction ($\beta$ = -0.314) and two coping strategies, that is resolution/acceptance ($\beta$ = -0.282) and denial ($\beta$ = 0.202), alterations in arousal and reactivity: by job satisfaction ($\beta$ = -0.294) resolution/acceptance ($\beta$ = -0.272) and regret ($\beta$ = 0.196).

## Discussion

Representatives of medical staff who participated in the study and who professionally provide help to sufferers show relatively high levels of secondary traumatic stress symptoms. It rendered higher than in PCL-5 standardization tests [63] with people who directly experienced various types of trauma and higher in relation to the representatives of other occupational groups which provide help for trauma victims such as therapists, social workers and probation officers [13]. As many as 43% of study participants present a high probability of the development of secondary posttraumatic stress disorder while in the group of therapists it amounted to only 7.5%.

The obtained results seem to show the medical personnel may be characterized by insufficient competences of coping with trauma experienced by others. The lack of sufficient competence to cope may also be related to the depletion of resources as a result of the high demands imposed by the poorly supportive environment and the burdensome working conditions [68, 69]. Ruotsalainen et al. [70] found that medical personnel may suffer from work-related stress as a result of lack of skills, low social support at work, and organisational factors, which can also result in inefficiency in dealing with trauma. This is an alarming phenomenon and it shows the need of including the medical staff in actions aimed at protection against the negative consequences of the experienced stress, especially through facilitating the development of trauma coping skills.

It should be underlined that the research conducted worldwide and in Poland confirm the high risk of secondary posttraumatic stress disorders among medical personnel, especially nurses [8, 9, 11–13]. It is of significance that in the case of medical staff—as opposed to other occupational groups whose members provide help for people who experienced trauma—the indirect exposure often coexists with direct traumatic experiences, including assault and aggression attacks from patients as well as other personal traumatic experiences. Interesting in this context is research conducted by Regehr et al. [71]. In this study higher levels of distress were found among paramedics who developed secondary trauma compared to experiencing

**Table 3. Regression analysis for variables predicting STS factors in the examined group of medical staff ($N = 419$).**

| Secondary Traumatic Stress factors | | | | | |
|---|---|---|---|---|---|
| **Predictors** | **B** | **$R^2$** | **$p$** | **F** | **$R^2$ for model** |
| **Intrusion** | | | | 23.200 | 0.312 |
| Resolution/acceptance | -0.236 | 0.13 | <0.001 | | |
| Job satisfaction | -0.223 | 0.08 | <0.001 | | |
| Regret | 0.208 | 0.04 | <0.001 | | |
| SS friends | -0.195 | 0.02 | <0.01 | | |
| Denial | 0.160 | 0.01 | <0.01 | | |
| Number of working hours | -0.158 | 0.01 | <0.05 | | |
| SS coworkers | 0.135 | 0.01 | <0.05 | | |
| Work experience | 0.119 | 0.01 | <0.05 | | |
| **Avoidance** | | | | 15.089 | 0.249 |
| Job satisfaction | -0.225 | 0.10 | <0.001 | | |
| Denial | 0.223 | 0.04 | <0.001 | | |
| Resolution/acceptance | -0.170 | 0.04 | <0.01 | | |
| Regret | 0.152 | 0.01 | <0.01 | | |
| SS friends | -0.142 | 0.01 | <0.05 | | |
| Number of working hours | -0.140 | 0.01 | <0.05 | | |
| SS coworkers | 0.127 | 0.01 | <0.05 | | |
| Work experience | 0.109 | 0.01 | <0.05 | | |
| Downward comparison | -0.100 | 0.01 | <0.05 | | |
| **Negative alterations in cognitions and mood** | | | | 38.995 | 0.321 |
| Job satisfaction | -0.314 | 0.16 | <0.001 | | |
| Resolution/acceptance | -0.282 | 0.06 | <0.001 | | |
| Denial | 0.202 | 0.06 | <0.001 | | |
| Regret | 0.196 | 0.03 | <0.01 | | |
| Number of working hours | -0.149 | 0.01 | <0.05 | | |
| **Alterations in arousal and reactivity** | | | | 28.749 | 0.258 |
| Job satisfaction | -0.294 | 0.14 | <0.001 | | |
| Resolution/acceptance | -0.272 | 0.05 | <0.001 | | |
| Regret | 0.196 | 0.04 | <0.01 | | |
| Denial | 0.131 | 0.01 | <0.05 | | |
| Number of working hours | -0.098 | 0.01 | <0.05 | | |

Abbreviations as in Table 2.

Cohen's $f^2 = 0.45$ (intrusion), 0.33 (avoidance), 0.47 (negative alterations in cognition and mood), 0.35 (alterations in arousal and reactivity).

direct trauma, as a result of working with a traumatized individual. The authors stressed that the empathetic relationship developed between the paramedic and the victim increases the vulnerability to experience an emotional response to the victim's suffering and develop symptoms of traumatic stress as a result.

A high risk of STSD occurrence among the medical personnel may result from the character of the performed work, everyday contact with suffering, pain, looking at mutilation, and death. This issue may be especially important in the present time of COVID-19 pandemic when stress connected with danger to one's health and life joins the regular stress factors related to providing help to the injured. During the pandemic, competence to deal with

traumatic situations (both one's own and others') effectively and large social support network are all the more desirable.

Among the analyzed variables the strongest relations with STS symptoms were presented by job satisfaction which seems to be playing a protective role as it prevents and alleviates the negative consequences of secondary exposure to trauma which are expressed in the form of STS symptoms. What is more, satisfaction with work was also the main predictor of symptoms included in all four STS dimensions. The obtained data confirm the results of research conducted among advisors working with substance abusers [50] in which job satisfaction mediated the relation between STS and engagement in work. It should be taken into account that low job satisfaction may be the reason for STS development. It is worth noting that the inverse relationship between variables is also possible. Some studies [52] found that secondary traumatic stress leads to job dissatisfaction. It means that the increase in STS may be accompanied by a decrease in job satisfaction.

The remaining work-related variables in the form of number of working hours per week and workload were negatively—although to a small extent—connected with STS symptoms in correlation analyses (it suggests that they may play a protective role). Taking into account regression analyses, number of working hours and work experience found to be predictors of STS; but weak and less relevant. It means that occupational load indicators play significantly lesser role in the occurrence of negative consequences of indirect trauma exposure. Possibly, the routine resulting from everyday contact with patients constitute a barrier against STS symptoms for the medical staff. It should be underlined that the data available in references do not provide a clear picture of interdependencies between the variables. This constitutes a need for further research in this area. According to some researchers [32, 72] the influence of occupational load on the consequences of secondary exposure to trauma is overemphasized. They underline that it is not the burden of working with traumatized people but rather qualifications, experience and training are the factors influencing the occurrence of negative consequences of secondary trauma exposure and if so that what will be their extent.

Among the four analyzed sources of support, the strongest links to STS symptoms are related to support obtained from family and friends than from supervisors and coworkers. This is rather consistent with the results of the mentioned research [57–59], but it stands in opposition to the results of the study conducted among medical personnel that indicates the relation of STS and social support obtained from the work environment [18, 31, 55]. The two source of social support play a predictive role for STS symptoms, i.e., friends and coworkers. It is interesting that social support from coworkers found to be a positive indicator of STS and may be identified as risk factor. However, the role of social support in STS prediction is negligible. The less significant role of social support for the STS symptoms may be connected with the character of the performed work. Other professionals working with trauma victims, especially therapists are supervised and use a full range of workshops and training aimed at increasing their competencies and developing resilience to stress. This is a rare case for medical personnel who is overworked. Social support may be significant in preventing other negative consequences of occupational stress, including burnout syndrome. Moreover, the dependence between social support and STS may have various forms. Support does not have to be directly connected to the increase in STS symptoms but it can mediate between secondary exposure to trauma and STS. In such a case it functions as a mediator.

Cognitive trauma processing, as results from the conducted study, is significant for the occurrence of STS symptoms, although its role turned out limited. The results of correlation analysis indicated that STS symptoms are positively connected with negative strategies and negatively with positive ones. Only the strategy of downward comparison is not significantly statistically connected with the STS symptoms. Resolution/acceptance, denial and regret (in

small extent) presented themselves as predictors of STS symptoms but their share in the explanation of the dependent variable is significantly lower than the share of job satisfaction. The regret strategy is connected with self-blame. This means that the specialists blame themselves for patient's pain and suffering which can significantly increase the susceptibility to the occurrence of secondary traumatic stress disorders. The strategy of denial through avoiding the processing of information related to client's trauma may be another STS risk factor. In turn, the rational attitude, that is the effort to solve the problem or accept the situation when solving it is impossible (resolution/acceptance), allows decreasing the level of STS symptoms.

The limited share of cognitive strategies of coping with trauma in the prediction of STS symptoms may result from e.g., stability of possessed cognitive schemes and weaker engagement of the medical personnel in the processing of trauma experienced by others. This would correspond with other data obtained by Michael et al. [10] which show that people facing direct threat reported more negative posttraumatic cognitions than those faced with an indirect threat.

## Limitations of the study

There are certain limitations to the conducted research. It was a cross-sectional study which does not allow to draw conclusions related to the cause and effect dependencies. Subjective indicators of indirect exposure to trauma e.g., in the form of evaluation of the size and meaning of the influence of the events experienced by patients treated as the severity of the perceived trauma were also not taken into account. The influence of personal traumatic experiences that could affect the level of STS symptoms was also not analyzed. The analyses did not include a certain place of work (hospital, ward) because of the possible simultaneous employment of study participants in multiple places. It should be underlined that the study group was not homogeneous. Men constituted the majority of the group of paramedics while the group of nurses included mainly women. Age, gender and occupational group were not taken into account in further analysis. Due to the complexity of the social support variable and possible overpowering of the study, the results should be interpreted with caution.

Despite the indicated limitations, the results of the conducted study contribute new information within the conditioning of negative consequences of indirect exposure to trauma among the medical personnel. It shows that the cognitive models developed for PTSD may be applied to STS. Moreover, the research in this topic available in the literature referring to the medical staff include first and foremost nurses, therefore, the additional advantage of the conducted study was the extension of the sample by the group of paramedics.

The conducted study may also inspire further research in which other indicators of cognitive trauma processing, such as disruptions in basic convictions or ruminating the events experienced by the patients as well as personal features of helpers including the feeling of self-sense of self-efficacy in coping with trauma experienced by others should also be included. The analyses indicating the mediational role of job satisfaction, social support, and cognitive trauma processing in the relationship between occupational load and STS symptoms would also be useful. Longitudinal study that allow capturing the changes in the range of STS symptoms is also necessary. It should be remembered that the indirect exposure to trauma leads not only to negative consequences but it also may be a source of positive posttraumatic changes in the form of vicarious growth after the trauma.

## Implications for practice

The conducted research may also have the practical implications for the development of prevention programs aiming at the decrease of levels of STS symptoms and lowering the risk of

STSD occurrence among medical personnel, nurses and paramedics. The procedures designed to increase the level of satisfaction with work seems to be important. These interventions should focus on improving the source of job satisfaction such as the perceived ability to deliver good patient care, good relationships, respect from the superiors, supportive leadership, good salary, competitive pay and bonuses, participation in developing own work schedule, job security, self-growth in the form of professional training and job promotion, job autonomy, opportunity to decision-making and develop multidisciplinary actions in the context of health [73–76]. Nikić et al. [77] also point out the need for improving the communication skills and health as interventions that may lead to increase job satisfaction among health care workers. It should also be taken into consideration that high job satisfaction may favour the occurrence of secondary posttraumatic growth (SPTG). This is indicated by the research of Ogińska-Bulik and Juczynski [13] which informs about a positive relationship between job satisfaction and SPTG in a group of therapists and nurses working with trauma victims. Moreover, the development of competences of coping with trauma considering the alteration of cognitive coping strategies from negative to positive as well as encouragement to use of various self-care practices is also advisable. The significance of such practices is mentioned by Molnar et al. [78]. Encouraging to search and use not only social support from the close ones but also various forms of support like participation in workshops, training, supervision, or debriefing is also recommended. According to Calderón-Abbo et al. [79] these forms of support may significantly contribute to the prevention of negative results of indirect exposure to trauma. A significant role for the reduction of STS symptoms is also attributed to psychoeducation aimed at providing and broadening the knowledge about STS and developing coping skills. Molnar et al. [78] underline their importance and efficiency in lowering the intensity of STS symptoms in pediatric nurses. It may also be useful for professional competences. Several studies mentioned by Molnar et al. [78] show that the development of such competencies contributed to the reduction of STS symptoms in professionals who work with trauma victims. *Med-Stress* program developed by Smoktunowicz et al. [80] and aimed at counteracting the occurrence of secondary traumatization among medical personnel is used in Poland.

## Conclusions

Paramedics and nurses are at the high risk of indirect traumatic exposure and thus may be more prone to secondary traumatic stress symptoms development. From all analyzed variables in the study, eight turned out to be the predictors of STS. The main predictor of STS symptoms was job satisfaction. The predictive role for STS was also demonstrated by two cognitive coping strategies i.e., regret (positive relation) and resolution/acceptance (negative relation). The contribution of other analyzed variables, i.e., denial, number of working hours, work experience, social support from friends and coworkers to explaining the dependent variable, was rather small. It is important to include the medical personnel in the actions aiming at prevention and reduction of STS symptoms.

## Supporting information

**S1 Table. STROBE checklist for cross-sectional studies.**
(DOC)

## Author Contributions

**Conceptualization:** Nina Ogińska-Bulik, Paulina Michalska.

**Data curation:** Nina Ogińska-Bulik, Piotr Jerzy Gurowiec, Paulina Michalska, Edyta Kędra.

**Formal analysis:** Nina Ogińska-Bulik, Piotr Jerzy Gurowiec, Paulina Michalska, Edyta Kędra.

**Funding acquisition:** Piotr Jerzy Gurowiec, Edyta Kędra.

**Investigation:** Nina Ogińska-Bulik, Paulina Michalska, Edyta Kędra.

**Methodology:** Nina Ogińska-Bulik, Paulina Michalska.

**Project administration:** Nina Ogińska-Bulik, Piotr Jerzy Gurowiec, Paulina Michalska, Edyta Kędra.

**Resources:** Nina Ogińska-Bulik, Piotr Jerzy Gurowiec, Paulina Michalska, Edyta Kędra.

**Software:** Nina Ogińska-Bulik, Piotr Jerzy Gurowiec, Paulina Michalska, Edyta Kędra.

**Supervision:** Nina Ogińska-Bulik, Piotr Jerzy Gurowiec, Paulina Michalska, Edyta Kędra.

**Validation:** Nina Ogińska-Bulik, Piotr Jerzy Gurowiec, Paulina Michalska, Edyta Kędra.

**Visualization:** Piotr Jerzy Gurowiec, Paulina Michalska.

**Writing – original draft:** Nina Ogińska-Bulik, Piotr Jerzy Gurowiec, Paulina Michalska, Edyta Kędra.

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
