## [Decision Letter · Decision Letter 0]

20 Oct 2020

PONE-D-20-24212

Prevalence and predictors of secondary traumatic stress disorder symptoms in health care professionals working with trauma victims: a cross-sectional study

PLOS ONE

Dear Dr. Michalska,

Thank you for submitting your manuscript to PLOS ONE. After careful consideration, we feel that it has merit but does not fully meet PLOS ONE’s publication criteria as it currently stands. Therefore, we invite you to submit a revised version of the manuscript that addresses the points raised during the review process.

In order to make the latter part of the introduction (from line 145) easier to understand, I recommend the authors to rewrite the part in the order shown below if possible:

(1) Explaining about Dutton and Rubinstein’s Ecological Framework of Trauma, especially each of four elements, taking into account the theory’s applicability to STS symptoms.

(2) Showing the reasons why the authors focused on the two of four elements, that is, coping strategies and environmental factors, as potential predictors of STS symptoms in this study. At that time, a detailed description of cognitive trauma theories (PTSD development model, emotional processing model, and Constructivist Self-development Theory) and their applicability to STS symptoms may be necessary.

(3) Reviewing previous studies that have examined the relationships between potential predictors (coping strategies and environmental factors) and STS symptoms; besides unsolved problems should be mentioned. A comprehensive review is not necessarily bad, but there is possibly no need to refer to the studies of PTSD symptoms.

(4) Describing the purpose and hypotheses of this study in the end of the introduction.

The authors should add necessary information as follows.

Sampling methods; *t*- and *p*-value if they carried out a one-sample *t*-test to examine the difference between the sample mean of STS (*M* = 31.0) and a particular value (*M* = 26.0), that is, another sample mean of PTSD among people who had experienced direct traumatic events; the total number of items of both the Job Satisfaction Scale and the Social Support Scale; the number of response options for the Social Support Scale; the number of items and Cronbach’s alpha of each of the five subscales of the Cognitive Processing of Trauma Scale; the effect size for the multiple regression analysis; and so on.

In addition, given that effect size = .30, *α* = .05, and *N* = 419, then power = .99 in the correlation analyses. The value of power indicates that this study is overpowered.

It is preferable that demographic variables such as gender, age, and occupation (nurse/paramedics) are entered as control variables at the first step in the multiple regression analysis if the authors intended to examine the relationships between the potential predictors and STS symptoms regardless of the demographic differences. Also, as the reviewer pointed out, please show all the independent variables initially entered in the multiple regression analysis. Furthermore, in the regression analysis, why did the authors consider only factors with contribution rate of more than 5% as the STS predictors? Please explain it clearly from the scientific viewpoint.

Finally, if the authors carried out the correlation analyses preliminarily, it seems better to mainly discuss the results of the multiple regression analysis.

We look forward to receiving your revised manuscript.

Kind regards,

Hirokazu TANIGUCHI, Ph.D.

Academic Editor

PLOS ONE

Journal Requirements:

2. Please provide additional details regarding participant consent. In the Methods section, please ensure that you have specified what type of consent you obtained (for instance, written or verbal) and whether the ethics committee approved this consent procedure. If verbal consent was obtained please state why it was not possible to obtain written consent and how verbal consent was recorded. If your study included minors, state whether you obtained consent from parents or guardians.

3. Please state specifically whether the IRB approved the study.

4. Please include your tables as part of your main manuscript and remove the individual files. Please note that supplementary tables should remain as separate "supporting information" files.

Reviewers' comments:

Reviewer's Responses to Questions

**Comments to the Author**

1. Is the manuscript technically sound, and do the data support the conclusions?

Reviewer #1: Partly

Reviewer #2: Partly

2. Has the statistical analysis been performed appropriately and rigorously? 

Reviewer #1: No

Reviewer #2: Yes

3. Have the authors made all data underlying the findings in their manuscript fully available?

Reviewer #1: Yes

Reviewer #2: Yes

4. Is the manuscript presented in an intelligible fashion and written in standard English?

Reviewer #1: Yes

Reviewer #2: No

5. Review Comments to the Author

Reviewer #1: As mentioned in L.200, I would like you to highlight the evidence that social support can be expected to be sufficiently effective for intense stress, such as trauma. Although social support is an important variable, it may not have been adequately controlled for in this study because there are so many different factors to consider, including the relationship with the person providing the support and the nature of the support.

As discussed in L.152, it is understandable that years of work experience and other factors may affect STS. However, even if the influence of work-related variables is found to be significant, it seems that work-related variables are difficult to control adequately. Please explain why you think it would be useful in supporting the nurses and others.

Please explain the method of selection of survey participants as described in L305.

The results presented in L. 397 are ambiguous as to whether a statistical test has been performed. Also, given that a variety of occupations are covered, is it appropriate to treat questions about occupational indicators as equal in quality?

I understand that job satisfaction is important, as stated in L518. If so, it would be good if you could mention the specific procedures for improving job satisfaction. In doing so, I think it is necessary to consider the characteristics of the occupation that was the subject of this study.

I think Limitation is adequately described.

Reviewer #2: Dear Authors,

Thank you for the opportunity to review this manuscript. It presents the results of a cross-sectional study on the prevalence and predictors of secondary traumatic stress resulting from professional care for trauma survivors. Since, according to data, traumatic experiences are common, the psychological cost of providing help is indeed an urgent issue. The study concerns healthcare professionals, which is all the more important as the COVID pandemic continues to spread. Hence, the choice of the subject is the strength of this manuscript. I would like to also highlight an informative and comprehensive Introduction section. And regarding some concerns that I list below, I hope you will find my suggestions helpful.

Abstract

1) I find the statement “In the explanation of STS symptoms occurrence two cognitive strategies are also applied” bit misleading (who applied what?). It would be beneficial to clarify this.

2) Also, I find it a bit difficult to follow what do you mean by “Paramedics and nurses are significantly exposed to the occurrence of secondary traumatic stress symptoms”. Do you mean: indirectly exposed to traumatic incidents or at high risk of developing secondary traumatic stress symptoms?

Introduction section

1) Clarification of why this particular set of potential STS predictors was chosen would be appreciated. Especially as you are exploring the effects of general risk factors such as workload as well as specific elements of traumatization mechanisms (i.e. cognitive processing of trauma).

2) It would be beneficial to distinguish between the terms: STS and STSD. In some parts of the manuscript, it seems like they are synonyms (e.g. “Exposure to indirect trauma may be connected with various mental health disorders, above all Secondary Traumatic Stress (STS), which is also described as Secondary Traumatic Stress Disorder). Later on, STSD is defined as a clinical manifestation of STS (above the diagnostic threshold, e.g. “ In turn, high levels of these symptoms signifying a high probability of STSD diagnosis occurred in 182 people which is 43.4% of study participants.”).

3) It's a bit confusing what theory was particularly adopted in the study. We can read that the research model refers to PTSD model by Ehlers and Clark, the Constructionist Self-development Theory, and Ecological Framework of Trauma by Dutton and Rubinstein.

4) I would suggest refraining from saying “assume” meaning “hypothesize”.

5) The manuscript would benefit from proofreading.

Materials and methods section

1) There is no information on how the sample size was determined. The N is quite big so the study might be “overpowered”.

2) Also, the recruitment details and sampling methods are not reported.

Results and Discussion Section:

1) It would be beneficial to indicate the initial set of variables entered in the regression model. Also, participants’ age was found as significantly related to STS. Thus, it is not clear whether it was included into the final analyses (as a control variable).

2) I would recommend shortening the correlation analysis report as correlations are preliminary analyses. The manuscript includes a related comprehensible table (Table 1).You state that “the increase of STS symptoms in the studied group of medical personnel who provides help for sufferers is higher than in standardization tests”, however it is not clear how to understand the increase of STS.

3) You indicate the role of job satisfaction as a significant predictor of STS. The results of the regression analysis do reflect this. However, as you pointed in the Limitations section, the study is cross-sectional, so it is difficult to decide what is the cause, and what is the effect. It is possible that while STS is rising, job satisfaction decreases. I would suggest putting more emphasis on this in the Discussion section. Especially since you state (in the Abstract, Results, and Conclusions sections) that "job satisfaction was the main predictor of STS symptoms," which points to the leading role of job satisfaction in predicting STS.

4) We can read that “occupational load indicators included in the research, i.e., the number of working hours per week and workload were negatively – although to a small extent – connected with STS symptoms”. However, it is not reflected in the regression results (Table 2).

5) You mention the importance of your results in the shade of COVID pandemic. Am I right, that the data was collected prior to the COVID-19 outbreak? If not, it would be important to at least discuss the its potential impact on medical personnel working conditions.

6) You state that “The obtained results point towards the fact that the medical personnel is characterized by insufficient competences of coping with trauma experienced by others”. Could you elaborate on that, as with this sentence alone it is ambiguous whether this conclusion is supported by the data.

Conclusions section

1) Same as in the Abstract, I find the statement “Paramedics and nurses are significantly exposed to the occurrence of secondary traumatic stress symptoms” bit confusing. Do you mean “indirectly exposed to trauma/traumatic incidents” or that they are at high risk of developing secondary traumatic stress symptoms?

2) The statement “Two of the three occupational load indicators found to be negatively related to STS” is not covered by the data (Table 2).

6. PLOS authors have the option to publish the peer review history of their article (what does this mean?). If published, this will include your full peer review and any attached files.

Reviewer #1: **Yes: **Shunsuke KOSEKI

Reviewer #2: No

---

## [Author Response · Author response to Decision Letter 0]

26 Nov 2020

In order to make the latter part of the introduction (from line 145) easier to understand, I recommend the authors to rewrite the part in the order shown below if possible:

(1) Explaining about Dutton and Rubinstein’s Ecological Framework of Trauma, especially each of four elements, taking into account the theory’s applicability to STS symptoms.

As the Editor suggested, we added the explanation of Dutton and Rubinstein’s Ecological Framework of Trauma where it was necessary and indicated by the Editor.

(2) Showing the reasons why the authors focused on the two of four elements, that is, coping strategies and environmental factors, as potential predictors of STS symptoms in this study. At that time, a detailed description of cognitive trauma theories (PTSD development model, emotional processing model, and Constructivist Self-development Theory) and their applicability to STS symptoms may be necessary.

As the Editor suggested, we tried to explain why we focused on the two of four elements, that was, coping strategies and environmental factors, as potential predictors of STS symptoms in this study. We added a sentence that showed applicability of PTSD models to STS symptoms.

(3) Reviewing previous studies that have examined the relationships between potential predictors (coping strategies and environmental factors) and STS symptoms; besides unsolved problems should be mentioned. A comprehensive review is not necessarily bad, but there is possibly no need to refer to the studies of PTSD symptoms.

As the Editor suggested, we put literature review after models description and resigned from reference to the studies of PTSD symptoms.

(4) Describing the purpose and hypotheses of this study in the end of the introduction.

As the Editor suggested, the purpose and hypotheses of this study were moved to the end of the Introduction section.

The authors should add necessary information as follows.

Sampling methods; t- and p-value if they carried out a one-sample t-test to examine the difference between the sample mean of STS (M = 31.0) and a particular value (M = 26.0), that is, another sample mean of PTSD among people who had experienced direct traumatic events; the total number of items of both the Job Satisfaction Scale and the Social Support Scale; the number of response options for the Social Support Scale; the number of items and Cronbach’s alpha of each of the five subscales of the Cognitive Processing of Trauma Scale; the effect size for the multiple regression analysis; and so on.

In addition, given that effect size = .30, α = .05, and N = 419, then power = .99 in the correlation analyses. The value of power indicates that this study is overpowered.

As the Editor suggested, the necessary information were added. 

It is preferable that demographic variables such as gender, age, and occupation (nurse/paramedics) are entered as control variables at the first step in the multiple regression analysis if the authors intended to examine the relationships between the potential predictors and STS symptoms regardless of the demographic differences. Also, as the reviewer pointed out, please show all the independent variables initially entered in the multiple regression analysis. Furthermore, in the regression analysis, why did the authors consider only factors with contribution rate of more than 5% as the STS predictors? Please explain it clearly from the scientific viewpoint.

As the Editor and Reviewer suggested, we indicated which variables were entered in the regression model. The gender, age and occupational group were not included at all as control variables because in regression analysis we mainly focused on factors stressed by Dutton and Rubinstein in the model of secondary traumatization, i.e., environmental, work-related variables and individual factors in the form of cognitive coping strategies. The occupational groups did not differ in the severity of STS symptoms. We put in into Limitation section. We hope it won't be a major constraint. Moreover, after consulting our statistics, we decided to abandon the assumption that only factors with contribution rate of more than 5% will be considered as the STS predictors. Despite the reference to literature, we found that this assumption may be incomprehensible to the reader due to different statistical models and approaches to science and statistics. We think it will be more suited to most interpretations. We hope that the Editor and Reviewers will share our opinion.

Finally, if the authors carried out the correlation analyses preliminarily, it seems better to mainly discuss the results of the multiple regression analysis.

We tried to shorten the discussion about correlation analysis in some places and highlight the results of the regression analysis, drawing attention to it.

Thank You so much for these valuable suggestions. We tried to respond to all comments.

Reviewer #1: As mentioned in L.200, I would like you to highlight the evidence that social support can be expected to be sufficiently effective for intense stress, such as trauma. Although social support is an important variable, it may not have been adequately controlled for in this study because there are so many different factors to consider, including the relationship with the person providing the support and the nature of the support.

We thank the Reviewer so much for that valuable concern. We agree with the Reviewer that controlling such a variable as social support is extremely difficult due to different types of support etc. In many places, we have tried to emphasize that we are exploring the relationship between STS and a specific source of support, which is also in line with the research conducted in this area (the authors also focused mainly on sources of support). Introducing additional distinctions could cause difficulties of interpretation. We will place this in the limitations of the study and in future studies we will try to control this variable in a more adequate way.

Manning-Jones, S., de Terte, I., & Stephens, C. (2016). Secondary traumatic stress, vicarious posttraumatic growth, and coping among health professionals; A comparison study. New Zealand Journal of Psychology, 45(1), 20–29.

As discussed in L.152, it is understandable that years of work experience and other factors may affect STS. However, even if the influence of work-related variables is found to be significant, it seems that work-related variables are difficult to control adequately. Please explain why you think it would be useful in supporting the nurses and others.

Of course, we try to explain it. So, in indirect occupational trauma, the degree of exposure is often expressed in terms of occupational load. Some authors indicate that occupational load, mainly in the form of a large number of clients/patients and a large amount of time spent working with them, is a major environmental risk factor for secondary traumatization. These variables are often omitted, or rather act as control, collateral variables, and studies cited in the literature review indicate important links between occupational load and STS. Moreover, in studies on the negative consequences of secondary trauma, it seems necessary to take into account work related variables because reducing occupational load is, among other things, a proven method of increasing job satisfaction, dealing with work stress, burnout or compassion fatigue. Moreover, the model of Dutton and Rubinstein stressed the role of environmental variables in the STS occurrence, also work-related variables.

Please explain the method of selection of survey participants as described in L305.

People, who agreed to participate in the study, were recruited from the medical personnel (nurses and paramedics) who worked in the centers where the study was conducted (hospitals and centers that have agreed to conduct psychological studies there). We managed to gain access to 430 respondents who met the inclusion criteria; the questionnaires were delivered to 430 people who had previously consented to participate in the study. The questionnaires were delivered to and collected from the respondents by the authors or persons trained by the authors during working hours. As a result, 419 people returned the questionnaires with fully data.

The results presented in L. 397 are ambiguous as to whether a statistical test has been performed. Also, given that a variety of occupations are covered, is it appropriate to treat questions about occupational indicators as equal in quality?

We added a statistic value to indicate the differences between samples. We understand that certain differences in the profession may generate problems when it comes to treating indicators as being of equal quality. However, in many studies conducted with the participation of different groups of professionals, the researchers also conducted analyses on the whole group of the surveyed persons (without specifying the professional group). So, suggesting literature data and this standard, we decided to do the same. We hope this will not introduce additional problems in data analysis. Moreover, both groups did not differ in the severity of STS symptoms.

Manning-Jones, S., de Terte, I., & Stephens, C. (2017). The Relationship Between Vicarious Posttraumatic Growth and Secondary Traumatic Stress Among Health Professionals. Journal of Loss and Trauma, 22(3), 256–270.

Manning-Jones, S., de Terte, I., & Stephens, C. (2016). Secondary traumatic stress, vicarious posttraumatic growth, and coping among health professionals; A comparison study. New Zealand Journal of Psychology, 45(1), 20–29.

I understand that job satisfaction is important, as stated in L518. If so, it would be good if you could mention the specific procedures for improving job satisfaction. In doing so, I think it is necessary to consider the characteristics of the occupation that was the subject of this study.

As the reviewer suggested, the procedures for improving job satisfaction were mentioned. We explained it in the section Implications for practice because we think that there it fits the best.

I think Limitation is adequately described.

Thank You so much for these valuable suggestions. We tried to respond to all comments.

Reviewer #2: Dear Authors,

Thank you for the opportunity to review this manuscript. It presents the results of a cross-sectional study on the prevalence and predictors of secondary traumatic stress resulting from professional care for trauma survivors. Since, according to data, traumatic experiences are common, the psychological cost of providing help is indeed an urgent issue. The study concerns healthcare professionals, which is all the more important as the COVID pandemic continues to spread. Hence, the choice of the subject is the strength of this manuscript. I would like to also highlight an informative and comprehensive Introduction section. And regarding some concerns that I list below, I hope you will find my suggestions helpful.

Thank You so much for these valuable suggestions. We tried to respond to all comments.

Abstract

1) I find the statement “In the explanation of STS symptoms occurrence two cognitive strategies are also applied” bit misleading (who applied what?). It would be beneficial to clarify this.

As the Reviewer suggested, this sentence was clarified.

2) Also, I find it a bit difficult to follow what do you mean by “Paramedics and nurses are significantly exposed to the occurrence of secondary traumatic stress symptoms”. Do you mean: indirectly exposed to traumatic incidents or at high risk of developing secondary traumatic stress symptoms?

As the Reviewer suggested, this sentence was rewritten for better understanding.

Introduction section

1) Clarification of why this particular set of potential STS predictors was chosen would be appreciated. Especially as you are exploring the effects of general risk factors such as workload as well as specific elements of traumatization mechanisms (i.e. cognitive processing of trauma).

We chose these predictors because the authors of Ecological Framework of Trauma (main base for our study) paid attention mostly to personal, environmental factors (connected with social support and work environment) and coping strategies. They explained that these factors are important because they influence the emotional reactions of those helping people working with trauma victims (reactions may be reflected as secondary traumatic stress), and coping strategies because coping activity is aimed to master the requirements of an individual's traumatic situation. Moreover, other authors, who see the applicability of PTSD models to STS symptoms, mainly stress the importance of cognitive factors, i.e. cognitive processing of trauma, which may take the form of trauma coping strategy, that is to say, it also refers to the ecological model.

2) It would be beneficial to distinguish between the terms: STS and STSD. In some parts of the manuscript, it seems like they are synonyms (e.g. “Exposure to indirect trauma may be connected with various mental health disorders, above all Secondary Traumatic Stress (STS), which is also described as Secondary Traumatic Stress Disorder). Later on, STSD is defined as a clinical manifestation of STS (above the diagnostic threshold, e.g. “ In turn, high levels of these symptoms signifying a high probability of STSD diagnosis occurred in 182 people which is 43.4% of study participants.”).

As the Reviewer suggested, we distinguished between STS and STSD. An additional explanation is that in relation to STS, the term severity of symptoms was used, while in relation to STSD it was referred to the risk of its occurrence. Moreover, the text uses the terms cited by the authors of the research we refer to. We hope that our explanation is satisfying for the Reviewer.

3) It's a bit confusing what theory was particularly adopted in the study. We can read that the research model refers to PTSD model by Ehlers and Clark, the Constructionist Self-development Theory, and Ecological Framework of Trauma by Dutton and Rubinstein.

We tried to simplify this paragraph by indicating the single model (Ecological Framework of Trauma by Dutton and Rubinstein) from which we drew the most when designing the study.

4) I would suggest refraining from saying “assume” meaning “hypothesize”.

As the Reviewer suggested, the word „assume” was replaced.

5) The manuscript would benefit from proofreading.

The manuscript was checked.

Materials and methods section

1) There is no information on how the sample size was determined. The N is quite big so the study might be “overpowered”.

In the research we aimed to obtain representativeness in order to be able to extrapolate the results to the target population. Despite some limitations in power statistics, it seems to us that the studies on large populations are a valuable trend, as the larger the sample, the greater the chance of being representative. Numerous attempts increase the chance of universality of obtained data. We hope that, maybe putting a sentence in the Limitations section about that the study should be interpreted with caution because it might be overpowered, will be sufficient.

2) Also, the recruitment details and sampling methods are not reported.

With regard to sampling methods, the sample included people who were exposed to secondary trauma (as defined in the inclusion criteria; purposive sampling). People, who agreed to participate in the study, were recruited from the medical personnel (nurses and paramedics) who worked in the centers where the study was conducted (hospitals and centers that have agreed to conduct psychological studies there). We managed to gain access to 430 respondents who met the inclusion criteria; the questionnaires were delivered to 430 people who had previously consented to participate in the study. The questionnaires were delivered to and collected from the respondents by the authors or persons trained by the authors during working hours. As a result, 419 people returned the questionnaires with fully data.

Results and Discussion Section:

1) It would be beneficial to indicate the initial set of variables entered in the regression model. Also, participants’ age was found as significantly related to STS. Thus, it is not clear whether it was included into the final analyses (as a control variable).

As the Reviewer suggested, we indicated which variables were entered in the regression model. The age was not included at all as control variable because in regression analysis we mainly focused on factors stressed by Dutton and Rubinstein in the model of secondary traumatization. We put it into Limitation section. We hope it won't be a major constraint.

2) I would recommend shortening the correlation analysis report as correlations are preliminary analyses. The manuscript includes a related comprehensible table (Table 1).You state that “the increase of STS symptoms in the studied group of medical personnel who provides help for sufferers is higher than in standardization tests”, however it is not clear how to understand the increase of STS.

As the Reviewer suggested, the correlation analysis description was shortened and the confusing statement was rewritten.

3) You indicate the role of job satisfaction as a significant predictor of STS. The results of the regression analysis do reflect this. However, as you pointed in the Limitations section, the study is cross-sectional, so it is difficult to decide what is the cause, and what is the effect. It is possible that while STS is rising, job satisfaction decreases. I would suggest putting more emphasis on this in the Discussion section. Especially since you state (in the Abstract, Results, and Conclusions sections) that "job satisfaction was the main predictor of STS symptoms," which points to the leading role of job satisfaction in predicting STS.

As the Reviewer suggested, we added some sentences that discussed the existing reverse relationship between STS and job satisfaction. We hope that it is sufficient for the Reviewer. 

4) We can read that “occupational load indicators included in the research, i.e., the number of working hours per week and workload were negatively – although to a small extent – connected with STS symptoms”. However, it is not reflected in the regression results (Table 2).

In this sentence we refer to the results of Table 1, where the results of the correlation analysis were presented. As for the correlations, workload and number of working hours found to be associated with STS. We apologize for misunderstanding and hope that now everything is clear.

5) You mention the importance of your results in the shade of COVID pandemic. Am I right, that the data was collected prior to the COVID-19 outbreak? If not, it would be important to at least discuss the its potential impact on medical personnel working conditions.

The research was conducted before the outbreak of the pandemic, but our aim was to indicate that during a pandemic, the consequences of exposure to stress for healthcare professionals may be greater. We explained why the results of the study may be important.

6) You state that “The obtained results point towards the fact that the medical personnel is characterized by insufficient competences of coping with trauma experienced by others”. Could you elaborate on that, as with this sentence alone it is ambiguous whether this conclusion is supported by the data.

As the Reviewer suggested, we added an explanation for this statement. 

Conclusions section

1) Same as in the Abstract, I find the statement “Paramedics and nurses are significantly exposed to the occurrence of secondary traumatic stress symptoms” bit confusing. Do you mean “indirectly exposed to trauma/traumatic incidents” or that they are at high risk of developing secondary traumatic stress symptoms?

As the Reviewer suggested, it was rewritten to better understanding.

2) The statement “Two of the three occupational load indicators found to be negatively related to STS” is not covered by the data (Table 2).

This statement regards the correlational analyses - number of working hours and workload found to be associated with STS. In fact, this paragraph was rewritten to avoid misunderstanding. We focused on results came from regression analysis.

---

## [Decision Letter · Decision Letter 1]

5 Jan 2021

PONE-D-20-24212R1

Prevalence and predictors of secondary traumatic stress symptoms in health care professionals working with trauma victims: a cross-sectional study

PLOS ONE

Dear Dr. Michalska,

Thank you for submitting your manuscript to PLOS ONE. After careful consideration, we feel that it has merit but does not fully meet PLOS ONE’s publication criteria as it currently stands. Therefore, we invite you to submit a revised version of the manuscript that addresses the points raised during the review process.

1) As the reviewer pointed out, the authors should explain the results of correlation analysis as briefly as possible.

2) I would advise using italics for letters used as statistical symbols or algebraic variables, such as *M*,* SD*,* n*, *R*^2^*, B*,* t*,* p*,* r*, *N*, *BE*,* F*,* f*^2^.

3) If there is no study or few studies of nurses in Poland which shows the relations between support from their friends and family and STS symptoms for now, the authors had better mention it after the sentence in L257.

4) The authors should add the reliability (Cronbach’s alpha) of the Job Satisfaction Scale.

5) I’d like to suggest that the authors move the sentence in L388 (A version adjusted for…) to before the sentence in L379 (The tool consists of …).

6) In Table 3, the beta value of job satisfaction for avoidance is -0.225. Is this correct? If so, denial should be placed above job satisfaction. 

Minor errors.

L41: Two positive cognitive strategies → Two cognitive strategies

L43: other analysed variables → other analyzed variables

L97: overtone that → overtone than

L368: range = 0 - 35 → range = 7 - 35

L384: -3 (I certainly disagree) → 0 (I certainly disagree)

L384: 3 (I certainly agree) → 6 (I certainly agree)

L444: *r* < 0.001 → *p* < 0.001

L459: variables explaining → variables explained

L493: denial (0.232) → denial (β = 0.232)

L587: pain and suffering what can → pain and suffering, which can

L636: developing own work scheduled → developing own work schedule

It seems preferable to revise as below:

L43: other analyzed variables, i.e., denial, workload….

L207: The negative connections of satisfaction from helping with STS symptoms, compassion fatigue, and burnout were presented in the study of American nurses.

L244: a significant role in alleviating STS symptoms.

L272: the occurrence of the negative relations of beliefs relating to both the goodness of the surrounding world and its comprehensibility to STS symptoms.

L419: *t *(417) = -1.336

L423: *t *(417) = -0.496

L426 (It would be better to rename several variables listed in Table 1 as follows): STS intrusion, STS avoidance, STS negative cognition and mood, STS arousal and reactivity, COPT positive cognitive restructuring, CPOT downward comparison, CPOT resolution/acceptance, CPOT denial, CPOT regret.

L580: The results of correlation analysis indicated that STS symptoms are….

L641: may favour the occurrence of secondary posttraumatic growth (SPTG).

L648: use not only social support from the close ones but also various forms of support….

L652: significant means for the reduction….

We look forward to receiving your revised manuscript.

Kind regards,

Hirokazu Taniguchi, Ph.D.

Academic Editor

PLOS ONE

Reviewers' comments:

Reviewer's Responses to Questions

**Comments to the Author**

1. If the authors have adequately addressed your comments raised in a previous round of review and you feel that this manuscript is now acceptable for publication, you may indicate that here to bypass the “Comments to the Author” section, enter your conflict of interest statement in the “Confidential to Editor” section, and submit your "Accept" recommendation.

Reviewer #1: All comments have been addressed

Reviewer #2: (No Response)

2. Is the manuscript technically sound, and do the data support the conclusions?

Reviewer #1: Yes

Reviewer #2: Partly

3. Has the statistical analysis been performed appropriately and rigorously? 

Reviewer #1: Yes

Reviewer #2: Yes

4. Have the authors made all data underlying the findings in their manuscript fully available?

Reviewer #1: Yes

Reviewer #2: Yes

5. Is the manuscript presented in an intelligible fashion and written in standard English?

Reviewer #1: Yes

Reviewer #2: Yes

6. Review Comments to the Author

Reviewer #1: Thank you for your appropriate revision. You have responded to all my comments appropriately. I wish you the best of luck in your research.

Reviewer #2: Dear Authors, thank you for the opportunity to read the revised version of this manuscript. The vast majority of suggestions have been included, thank you for that. This version is precise and clear. For further processing, I suggest again not to include a detailed description of the correlation analysis. In this case, nothing has changed compared to the original manuscript: the correlation analysis (excluding Table 1) accounts for almost half of the content of the Results section. However, this is a preliminary analysis, and the regression is the core one. Table 1 is self-explainable though.

I also noticed some inconsistencies with the references. In line 639 you refer to: Dragana et al. [77] but there is no such reference on the list (77. Stanković A, Nikolic M, Nikić D, Arandjelović M. Job satisfaction in health care workers. Acta Medica Medianae. 2008; 47:9-12).

I hope you find these suggestions useful for your valuable manuscript.

7. PLOS authors have the option to publish the peer review history of their article (what does this mean?). If published, this will include your full peer review and any attached files.

Reviewer #1: **Yes: **Shunsuke Koseki

Reviewer #2: No

---

## [Author Response · Author response to Decision Letter 1]

16 Jan 2021

Editor:

Thank You so much for the suggestions.

1) As the reviewer pointed out, the authors should explain the results of correlation analysis as briefly as possible.

As the Editor suggested, it was explained as briefly as possible.

2) I would advise using italics for letters used as statistical symbols or algebraic variables, such as M, SD, n, R2, B, t, p, r, N, BE, F, f2.

As the Editor suggested, the edit was done. 

3) If there is no study or few studies of nurses in Poland which shows the relations between support from their friends and family and STS symptoms for now, the authors had better mention it after the sentence in L257.

As the Editor suggested, the edit was done. 

4) The authors should add the reliability (Cronbach’s alpha) of the Job Satisfaction Scale.

As the Editor suggested, the reliability (Cronbach’s alpha) of the Job Satisfaction Scale was added.

5) I’d like to suggest that the authors move the sentence in L388 (A version adjusted for…) to before the sentence in L379 (The tool consists of …).

As the Editor suggested, the edit was done. 

6) In Table 3, the beta value of job satisfaction for avoidance is -0.225. Is this correct? If so, denial should be placed above job satisfaction. 

As the Editor suggested, it was corrected. 

Minor errors.

L41: Two positive cognitive strategies → Two cognitive strategies

As the Editor suggested, the edit was done. 

L43: other analysed variables → other analyzed variables

As the Editor suggested, the edit was done. 

L97: overtone that → overtone than

As the Editor suggested, the edit was done. 

L368: range = 0 - 35 → range = 7 – 35

As the Editor suggested, the edit was done. 

L384: -3 (I certainly disagree) → 0 (I certainly disagree)

As the Editor suggested, the edit was done. 

L384: 3 (I certainly agree) → 6 (I certainly agree)

As the Editor suggested, the edit was done. 

L444: r < 0.001 → p < 0.001

As the Editor suggested, the edit was done. 

L459: variables explaining → variables explained

As the Editor suggested, the edit was done. 

L493: denial (0.232) → denial (β = 0.232)

As the Editor suggested, the edit was done. 

L587: pain and suffering what can → pain and suffering, which can

As the Editor suggested, the edit was done. 

L636: developing own work scheduled → developing own work schedule

As the Editor suggested, the edit was done. 

It seems preferable to revise as below:

L43: other analyzed variables, i.e., denial, workload….

As the Editor suggested, the edit was done. 

L207: The negative connections of satisfaction from helping with STS symptoms, compassion fatigue, and burnout were presented in the study of American nurses.

As the Editor suggested, the edit was done. 

L244: a significant role in alleviating STS symptoms.

As the Editor suggested, the edit was done. 

L272: the occurrence of the negative relations of beliefs relating to both the goodness of the surrounding world and its comprehensibility to STS symptoms.

As the Editor suggested, the edit was done. 

L419: t (417) = -1.336

As the Editor suggested, the edit was done. 

L423: t (417) = -0.496

As the Editor suggested, the edit was done. 

L426 (It would be better to rename several variables listed in Table 1 as follows): STS intrusion, STS avoidance, STS negative cognition and mood, STS arousal and reactivity, COPT positive cognitive restructuring, CPOT downward comparison, CPOT resolution/acceptance, CPOT denial, CPOT regret.

As the Editor suggested, the edit was done. 

L580: The results of correlation analysis indicated that STS symptoms are….

As the Editor suggested, the edit was done. 

L641: may favour the occurrence of secondary posttraumatic growth (SPTG).

As the Editor suggested, the edit was done. 

L648: use not only social support from the close ones but also various forms of support….

As the Editor suggested, the edit was done. 

L652: significant means for the reduction….

As the Editor suggested, the edit was done. 

Reviewers:

Reviewer #1: Thank you for your appropriate revision. You have responded to all my comments appropriately. I wish you the best of luck in your research.

Thank You so much. 

Reviewer #2: Dear Authors, thank you for the opportunity to read the revised version of this manuscript. The vast majority of suggestions have been included, thank you for that. This version is precise and clear. For further processing, I suggest again not to include a detailed description of the correlation analysis. In this case, nothing has changed compared to the original manuscript: the correlation analysis (excluding Table 1) accounts for almost half of the content of the Results section. However, this is a preliminary analysis, and the regression is the core one. Table 1 is self-explainable though.

As the Reviewer suggested, the Results section regarding correlational analysis was explained as briefly as possible. 

I also noticed some inconsistencies with the references. In line 639 you refer to: Dragana et al. [77] but there is no such reference on the list (77. Stanković A, Nikolic M, Nikić D, Arandjelović M. Job satisfaction in health care workers. Acta Medica Medianae. 2008; 47:9-12).

Thank You so much for this comment. It was a little mistake in the author’s name. It was corrected. 

I hope you find these suggestions useful for your valuable manuscript.

Thank You for the suggestions. They were very useful.

---

## [Decision Letter · Decision Letter 2]

10 Feb 2021

Prevalence and predictors of secondary traumatic stress symptoms in health care professionals working with trauma victims: a cross-sectional study

PONE-D-20-24212R2

Dear Dr. Michalska,

We’re pleased to inform you that your manuscript has been judged scientifically suitable for publication and will be formally accepted for publication once it meets all outstanding technical requirements.

Kind regards,

Hirokazu Taniguchi, Ph.D.

Academic Editor

PLOS ONE

Additional Editor Comments (optional):

Reviewers' comments:

Reviewer's Responses to Questions

**Comments to the Author**

1. If the authors have adequately addressed your comments raised in a previous round of review and you feel that this manuscript is now acceptable for publication, you may indicate that here to bypass the “Comments to the Author” section, enter your conflict of interest statement in the “Confidential to Editor” section, and submit your "Accept" recommendation.

Reviewer #1: All comments have been addressed

Reviewer #2: All comments have been addressed

2. Is the manuscript technically sound, and do the data support the conclusions?

Reviewer #1: Yes

Reviewer #2: Yes

3. Has the statistical analysis been performed appropriately and rigorously? 

Reviewer #1: Yes

Reviewer #2: Yes

4. Have the authors made all data underlying the findings in their manuscript fully available?

Reviewer #1: Yes

Reviewer #2: Yes

5. Is the manuscript presented in an intelligible fashion and written in standard English?

Reviewer #1: Yes

Reviewer #2: Yes

6. Review Comments to the Author

Reviewer #1: I have determined that all comments have been properly corrected. I hope that the publication of this paper will lead to the development of research in this area.

Reviewer #2: Thank you for responding to all comments, I have no further notes. I wish you all the best with further proceedings.

7. PLOS authors have the option to publish the peer review history of their article (what does this mean?). If published, this will include your full peer review and any attached files.

Reviewer #1: **Yes: **Shunsuke Koseki

Reviewer #2: No

---

## [Editor Report · Acceptance letter]

12 Feb 2021

PONE-D-20-24212R2 

 Prevalence and predictors of secondary traumatic stress symptoms in health care professionals working with trauma victims: a cross-sectional study 

Dear Dr. Michalska:

I'm pleased to inform you that your manuscript has been deemed suitable for publication in PLOS ONE. Congratulations! Your manuscript is now with our production department. 

Kind regards, 

on behalf of

Dr. Hirokazu Taniguchi 

Academic Editor

PLOS ONE